# Protective Effect of *Cimicifuga racemosa* (L.) Nutt Extract on Oocyte and Follicle Toxicity Induced by Doxorubicin during In Vitro Culture of Mice Ovaries

**DOI:** 10.3390/ani13010018

**Published:** 2022-12-20

**Authors:** Ernando I. T. de Assis, Venância A. N. Azevedo, Miguel F. De Lima Neto, Francisco C. Costa, Laís R. F. M. Paulino, Pedro A. A. Barroso, Mariana A. M. Donato, Christina A. Peixoto, Alane P. O. Do Monte, Maria H. T. Matos, Alana N. Godinho, Jordânia M. O. Freire, Ana L. P. S. Batista, José R. V. Silva, Anderson W. B. Silva

**Affiliations:** 1Laboratory of Biotechnology and Physiology of Reproduction (LABIREP), Federal University of Ceara, Av. Maurocélio Rocha Ponte 100, Sobral 62041-040, CE, Brazil; 2Nucleus of Research in Animal Experimentation (NUPEX), Federal University of Ceara, Av. Maurocélio Rocha Ponte 100, Sobral 62041-040, CE, Brazil; 3Laboratory of Ultrastructure, CNPqAM/FIOCRUZ, Federal University of Pernambuco, Recife 50670-901, PE, Brazil; 4Nucleus of Biotechnology Applied to Ovarian Follicle Development, Federal University of São Francisco Valley, Rodovia BR 407, Km 12, Lote 543—Projeto de Irrigação Nilo Coelho—S/N, Petrolina 56300-990, PE, Brazil

**Keywords:** folliculogenesis, in vitro culture, chemotherapy, fertility preservation, phototherapy

## Abstract

**Simple Summary:**

Doxorubicin (DOXO) is a chemotherapeutic drug that promotes the loss of follicular reserve through atresia and overactivation of primordial follicles and, consequently, decrease fertility in female patients. Gonadotoxic protective agents that act independently of chemotherapy drug have great value for human medicine. *Cimicifuga racemosa* (L.) Nutt extract (CIMI) improves antioxidant status and has the potential to prevent formation of reactive oxygen species (ROS) in the ovary. However, there are no reports about the potential of CIMI to protect oocytes and follicles against damages caused by DOXO in ovarian tissues cultured in vitro. This study evaluated the potential of CIMI extract to reduce the deleterious effects of DOXO in oocytes, follicles and stromal cells in mice ovaries cultured in vitro. The results show that DOXO reduces the percentage of normal follicles and the density of stromal cells in cultured ovaries, but these harmful effects were blocked by CIMI. Higher staining intensity for caspase-3 was seen in ovaries cultured in control medium alone or with DOXO when compared with those cultured with CIMI alone or both CIMI and DOXO. Furthermore, ovaries cultured with CIMI had higher levels of mRNA for antioxidant enzymes, such as superoxide dismutase (SOD) and catalase (CAT).

**Abstract:**

This study evaluated the potential of *Cimicifuga racemosa* (L.) Nutt extract (CIMI) to reduce the deleterious effects of doxorubicin (DOXO) in oocytes, follicles and stromal cells in mice ovaries cultured in vitro. In experiment 1, mice ovaries were cultured in DMEM^+^ alone or supplemented with 5, 50 or 500 ng/mL CIMI, while in experiment 2, mice ovaries were cultured in DMEM^+^ alone or supplemented with 5 ng/mL CIMI (better concentration), 0.3 μg/mL DOXO or both. Thereafter, the ovaries were processed for histological (morphology, growth, activation, extracellular matrix configuration and stromal cell density), immunohistochemical (caspase-3) analyses. Follicle viability was evaluated by fluorescence microscopy (ethidium homodimer-1 and calcein) while real-time PCR was performed to analyses the levels of (mRNA for SOD, CAT and *nuclear factor erythroid 2–related factor 2* (NRF2) analyses. The results showed that DOXO reduces the percentage of normal follicles and the density of stromal cells in cultured ovaries, but these harmful effects were blocked by CIMI. The DOXO reduced the percentage of primordial follicles, while the presence of CIMI alone did not influence percentage of primordial follicles. A higher staining for caspase-3 was seen in ovaries cultured in control medium alone or with DOXO when compared with those cultured with CIMI alone or both CIMI and DOXO. In addition, follicles from ovaries cultured with both CIMI and DOXO were stained by calcein, while those follicles cultured with only DOXO were stained with ethidium homodimer-1. Furthermore, ovaries cultured with CIMI or both CIMI and DOXO had higher levels of mRNA for SOD and CAT, respectively, than those cultured with only DOXO. In conclusion, the extract of CIMI protects the ovaries against deleterious effects of DOXO on follicular survival and ovarian stromal cells.

## 1. Introduction

It is well documented that women are 38% less likely to become pregnant after cancer diagnosis and treatment [1]. Cancer-related infertility depends mainly on gonadal toxicity and loss of reproductive function in young women [2]. One of the main chemotherapeutic drugs known to decrease fertility in female patients is doxorubicin (DOXO), an anthracycline antibiotic, which mainly acts by interfering with the synthesis of nucleic acids through a process called intercalation, which prevents the duplication and separation of chains of DNA and RNA. This process induces the production of reactive oxygen species (ROS), blocks protein synthesis, promotes oxidative damage and triggers cell death by apoptosis [3]. DOXO is one of the most effective anticancer drugs and it is widely used in the treatment of various human tumors, such as neurofibromatosis, liver cancer, and breast cancer [4]. DOXO, however, promotes loss of follicular reserve through atresia and over activation of primordial follicles [5] and deletion of secondary follicles [6]. In this sense, oncofertility is in constant progress and evolves to develop new options for preserving fertility [7].

Gonadotoxic protective agents that act independently of chemotherapy drug have great value for human medicine. They act by blocking a common pathway of damage or by reducing the release of drugs to the ovary [1]. Plant extracts have several active compounds, such as antioxidants, which are reported as attractive alternatives to protect the ovary against a specific chemotherapy or class of drugs [8]. In a recent study, rutin, a powerful natural antioxidant, was able to maintain mitochondrial activity and to protect mouse ovarian follicles from oxidative damage caused by cisplatin [9]. In addition, polyphenols, such as curcumin, gallate, epicatechin and resveratrol, have potent anti-tumor properties and protect reproductive system of cancer patients [10].

*Cimicifuga racemosa* (L.) Nutt., (Black Cohosh; *Cimicifuga racemosa*) (CIMI) is a plant belonging to the Ranunculaceae family. The root of CIMI consists of more than 42 triterpene glycosides, 11 phenolic acids and more than 70 alkaloids and tannins. This plant has been traditionally used in China, Europe and North America for centuries to treat a variety of diseases, including symptoms of menopause and osteoporosis [11]. CIMI is used in treatment of menopausal symptoms, and exhibits antioxidant, anti-inflammatory, antidiabetic, antiviral, antiangiogenic, vasodilatory and immunosuppressive effects [12]. In vivo, CIMI extract improved antioxidant status, hormonal and lipid profile, glucose levels and liver functions in rats with polycystic ovary syndrome (PCOS), besides inducing the expression of Ki-67 in granulosa, theca and stromal cells. Expression of Ki67 is an indicator of cellular proliferation and its increase is indicative of high androgen production in ovaries of patients with PCOS [13]. When tested in vitro, CIMI extract preserved mitochondrial integrity and ATP levels, whereas prevented the formation of mitochondrial ROS, loss of mitochondrial membrane potential and cell death after oxidative stress induced by erastin and RSL-3 in neuronal and liver cells [14]. In this sense, CIMI extract can be able to reduce the damage in the ovary caused by DOXO through its antioxidant action. However, there are no reports about the potential of CIMI to protect ovaries against damages caused by DOXO, as well as on its mechanisms of action in ovarian tissues cultured in vitro.

The aim of this study was to investigate the effects of different concentration of CIMI on mice ovarian tissue cultured in vitro, and to evaluate if CIMI prevents the ovarian damage caused by DOXO in cultured ovaries. In addition, the effects of this extract on follicular morphology, activation of primordial follicles, maintenance of ovarian stromal cells density, configuration of extracellular matrix configuration (ECM), and follicular viability were evaluated.

## 2. Material and Methods

### 2.1. Chemicals

Doxorubicin (CAS number: 25316-40-9, Sigma), *Cimicifuga racemosa* extract (CAS number: 8477626-1, Sigma), the culture medium and other chemicals used in this study were purchased from Sigma Chemical Co. (St. Louis, MO, USA), unless otherwise indicated in the text. *Cimicifuga racemosa* powered extract was dissolved in 60% (*v*/*v*) ethanol for all experiments [14].

### 2.2. Animals and Evaluation of Estrous Cycle

This study was conducted in accordance with the guidelines and normative resolutions of the National Council for Control in Animal Experimentation (CONCEA) and the ethical guidelines from Ethics Committee on the Use of Animals (CEUA) of Federal University Ceará (approved under protocol No. 05/18). Adult Swiss mice (*Mus musculus*) were kept in polyethylene boxes lined with wood shavings (6 animals/box), with free access to filtered water and feed. The animals were kept at an average temperature of 22 ± 2 °C, following 12 h light-dark cycles.

All female mice from 18 g and/or 2 months of age had their estrous cycle evaluated once a day, for 15 days, between 9:00 h and 10:00 h a.m., by a single evaluator, as established by Marcondes [15]. According to the observed cells, the cycle stage, i.e., proestrus, estrus, metestrus or diestrus, was determined. Only females with a regular cycle, that lasts 4 to 5 days, were used to carry out the experiment.

### 2.3. Experiment 1: Effects of Different Concentration of CIMI on Follicle Morphology

The ovaries (n = 36) from mice (n = 18) with 18 g and/or 2 months old were cultured according to the protocol described by O’Brien [16]. In experiment 1, the ovaries were collected using surgical scissors and 28 G needles and cultured individually in 24-well plates containing 1 mL of DMEM/HAMS F12 supplemented with ascorbic acid (10 μg/mL), penicillin G (75 µg/mL), Insulin-Transferrin-Selenium (ITS) (10 μg/mL insulin, 5.5 μg/mL transferrin, and 5 ng/mL selenium) and Bovine Serum Albumin (BSA) (10 μg/mL). This base medium was called DMEM^+^. Next, the ovaries were cultured in DMEM^+^ alone or supplemented with 5, 50 or 500 ng/mL CIMI, and cultured at 37.5 °C, in 5% CO_2_, for 6 days. This dose-response curve was established to define the better concentration of CIMI to be used in experiment 2. Half of the culture medium was replaced every 2 days. Uncultured (fresh control) or cultured ovaries were fixed in paraformaldehyde and used for analysis of survival, activation and follicular growth, as well as configuration of the ECM. After obtaining the results of this experiment, the better concentration of CIMI extract was identified and tested to assess its potential to block the deleterious effects of DOXO in cultured ovaries.

### 2.4. Morphological Assessment of Ovarian Follicles and Evaluation of Cell Density

Uncultured and cultured ovaries were fixed in paraformaldehyde (4% in phosphate buffered saline—PBS, pH 7.4) for 24 h and processed for classical histology. After fixation, the ovaries were dehydrated in gradual series of ethanol, clarified with xylene and embedded in paraffin wax. For each ovary, 7 µm sections were mounted on slides and stained using the Hematoxylin-Eosin (HE) method. Quantitative analysis of ovarian sections was carried out by an experienced researcher who is unaware of the treatments of each group under analysis. The follicles were classified as primordial or growing follicles (primary and secondary follicles) according to Lins et al. [9]. These follicles were individually classified as morphologically normal when an intact oocyte was surrounded by granulosa cells well organized in 1 or more layers, and which had no pycnotic nucleus. Degenerated follicles were defined as those with retracted oocyte, with pycnotic nucleus and/or surrounded by disorganized granulosa cells, detached from the basement membrane [17]. Overall, 130 follicles were evaluated for each treatment.

To evaluate cellular density in the ovaries, the number of stromal cells was calculated in an area of 100 μm^2^. The average number of stromal cells per field was calculated as previously described [18]. All evaluations and measurements were performed by a single operator.

### 2.5. Analysis of Extracellular Matrix

To assess collagen fibers of extracellular matrix of ovarian cortex, staining with Picrosirius Red (Abcam Kit) was performed following the methodology described by Rittié [19] with modifications. Ovarian sections of 7 µm were dewaxed in xylene and incubated in Sirius Red solution (0.1%) for 1 h at room temperature. Next, the excess dye was removed with acetic acid solution (0.5%) and the sections were dehydrated and subjected to slide assembly with subsequent observation under an optical microscope (Nikon, Eclipse, TS 100, Tokyo, Japan) under magnification 400×. For each treatment, the percentage of the area occupied by collagen fibers in ten different fields was measured with the aid of a DS Cooled DS DS-Ri1 camera attached to a microscope (Nikon, Eclipse, TS 100, Tokyo, Japan), the microscope and the images were analyzed by Image J Software (Version 1.51p, 2017) with 400× magnification. Only the collagen fibers were marked in red with the picrosirius color, while the follicles remained colorless (white). The analyzer software automatically excludes the circumference of unstained follicles from the total area marked in red. The Image J Software was used to quantify the percentage of collagen fiber in uncultured and cultured tissues. The staining intensity of collagen fibers was determined by measuring the average pixel intensity of the total area imaged after background subtraction.

### 2.6. Experiment 2: Effects of CIMI and DOXO on Follicle Morphology, Viability and Gene Expression

Female mice (n = 16) after euthanasia had their ovaries (n = 32) collected and cultured in (i) DMEM^+^ alone or supplemented with 5 ng/mL CIMI extract, 0.3 μg/mL DOXO or both 5 ng/mL CIMI extract and 0.3 μg/mL DOXO (CIMI + DOXO). The in vitro culture was performed at 37.5 °C, in 5% CO_2_ for 6 days. Half of the culture medium was replaced every 2 days. After culture, histological analysis was performed to evaluate follicular survival, activation, growth, and viability, as well as ECM configuration, and stromal cells density as described in experiment 1. Moreover, the protein expression for caspase-3 was evaluated by immunohistochemistry.

### 2.7. Analysis of Follicular Viability after Ovarian Culture

To evaluate follicular viability, follicles at different stages of development were mechanically isolated using 26-G needles from cultured ovaries (five ovaries per treatment). After isolation, the follicles were transferred to 100 µL drops of DMEM^+^ and incubated with 4 mM of calcein-AM and 2 mM homodimer ethidium-1 (Molecular Probes, Invitrogen, Karlsruhe, Germany) at 37 °C for 15 min and, finally, the follicles were examined under a fluorescence microscope. The fluorescent signals emitted by calcein-AM and homodimer ethidium-1 were collected at 488 and 568 nm, respectively. Granulosa cells and oocytes were considered alive when the cytoplasm was positively marked with calcein-AM (green) and if the chromatin was not marked with homodimer-ethidium-1 (red) [20]. On the other hand, cells with their chromatin labeled with ethidium homodimer-1 (red) were considered non-viable.

### 2.8. Immunohistochemistry

Immunohistochemistry was performed as previously described by Barberino [21]. Ovaries were collected from the mice and fixed in fixed in paraformaldehyde (4%). The ovarian tissue was dehydrated with increasing concentrations of ethanol (Dinamica), clarified in xylene (Dinamica), and embedded in paraffin (Dinamica). The ovarian sections (5 μm thick) were mounted on Starfrost glass slides (Knittel, Braunschweig, Germany). The slides were incubated in citrate buffer (Dinamica) at 95 °C in a decloaking chamber (Biocare, Concord, CA, USA) for 40 min to retrieve antigenicity, and endogenous peroxidase activity was prevented by incubation with 3% H_2_O_2_ (Dinamica) and methyl ethanol (QEEL, São Paulo, Brazil) for 10 min. Nonspecific binding sites were blocked using 1% normal goat serum (Biocare) and diluted in phosphate-buffered saline (PBS; Sigma Aldrich Chemical Co., St. Louis, MO, USA). Subsequently, the sections were incubated in a humidified chamber for 90 min at room temperature with a primary antibody directed against activated caspase-3 (Anti-Caspase-3 antibody (ab49822, ABCAM) diluted in TBS, enriched with 0.3% (*v*/*v*) Tx and 5% (*w*/*v*) BSA (1/225). The specificity of this antibody has already been evaluated in previous studies with mice [22]. Thereafter, the sections were incubated for 30 min with MACH4 Universal HRP-polymer (Biocare). Protein localization was demonstrated with diaminobenzidine (DAB; Biocare), and the sections were counterstained with hematoxylin (Vetec, Sao Paulo, Brazil) for 1 min. Negative controls (reaction control) were performed by replacing the primary antibody directed against activated caspase-3 by normal rabbit IgG. Using an optical microscope (Nikon) connected to a computer equipped with Image-Pro Plus^®^ software (Media Cybernetics) five photos were taken at the same magnification (100×) and analyzed using the Gimp 2.6 software program (GNU Manipulation Program images, UNIX platforms).

### 2.9. Ultrastructural Analysis

Ovarian fragments were fixed overnight in a solution containing 2.5% glutaraldehyde and 4% paraformaldehyde in 0.1 M cacodylate buffer. After fixation, the samples were washed twice in the same buffer and post-fixed in a solution containing 1% osmium tetroxide, 2 mM calcium chloride and 0.8% potassium ferricyanide in 0.1 M cacodylate buffer, pH 7.2, dehydrated in acetone and embedded in Embed 812. Polymerization was performed at 60 °C for 3 days. Ultrathin sections were collected on 300-mesh nickel grids, counterstained with 5% uranyl acetate and lead citrate and examined using a FEI Morgani 268D transmission electron microscope [23].

### 2.10. RNA Isolation and Real Time Quantitative PCR (qPCR)

After 6 days of culture, the ovaries of each treatment were collected and stored at −80 °C until total RNA extraction for further analysis of mRNA levels for SOD, CAT and NRF2. Total RNA extraction was performed using a TRIzol^®^ purification kit (Invitrogen, São Paulo, Brazil) according to the manufacturer’s instructions. Reverse transcription was performed in a total volume of 20 μL composed of 10 μL of sample containing 1 mg of RNA, 4 μL reverse transcriptase buffer (Invitrogen), eight units RNAsin, 150 units of reverse transcriptase Super- script III, 0.036 U random primers, 10 mM dithiothreitol and 0.5 mM of each dNTP (Invitrogen). The mixture was incubated at 42.1 °C for 1 h, subsequently at 80 °C for 5 min, and finally stored at −20 °C. The negative control was prepared under the same conditions but without the addition of reverse transcriptase. The mRNA quantification was performed using SYBR Green. The qPCR reactions were composed of 1 μL of cDNA as a template in 9.4 μL of SYBR Green Master Mix (PE Applied Biosystems, Foster City, CA, USA), 9.4 μL of ultra-pure water and 0.5 μM of each primer. The primers were designed to perform amplification of SOD, CAT, NRF2 and glyceraldehyde-3-phosphate dehydrogenase (GAPDH) (Table 1). GAPDH was used as the reference gene. The specificity of each primer pair was confirmed using melting curve analysis of qPCR products. The thermal cycling profile for the first round of qPCR was initial denaturation and polymerase activation for 10 min at 95 °C, followed by 40 cycles of 15 s at 95 °C, 30 s at 58 °C and 30 s at 72 °C. The final extension was for 10 min at 72 °C. All reactions were performed on a Step One Plus instrument (Applied Biosystems, Foster City, CA, USA). The ΔΔCt method was used to transform Ct values into normalized values for relative expression levels.

### 2.11. Statistical Analysis

The statistical analysis was performed using the software GraphPad Prism. The percentages of normal follicles, as well those of primordial and developing follicles were evaluated by Chi-square test. Data of collagen fibers distribution, stromal cell density, levels of mRNA for SOD, CAT and NRF2 were analyzed by the Kruskal-Wallis test, followed by Dunn’s comparison. The results were expressed as mean and standard error (mean ± S.E.M). Differences were considered significant when *p* < 0.05.

## 3. Results

### 3.1. Experiment 1: Effect of CIMI on Follicular Morphology, Activation and Development after In Vitro Culture

Ovaries cultured in medium supplemented with 5, 50 or 500 ng/mL CIMI had higher (*p* < 0.05) percentages of morphological normal follicles than those cultured in control medium (DMEM^+^) (*p* > 0.05) (Figure 1A). The morphology of follicles from ovaries cultured with DMEM^+^ alone or supplemented with CIMI is shown in Figure 1B,C.

The presence of 5, 50 or 500 ng/mL CIMI did not influence the percentages of primordial and developing follicles (Figure 2A,B) when compared to ovaries cultured with DMEM^+^ alone (*p* < 0.05).

### 3.2. Evaluation of Ovarian Extracellular Matrix after In Vitro Culture

Ovaries cultured in medium containing 50 ng/mL CIMI had reduced percentage of collagen fibers when compared with those cultured with only DMEM^+^ (Figure 3). Nevertheless, when ovaries were cultured in the presence of 5 or 500 ng/mL CIMI, the tissues maintained their percentage of collagen fibers similar to those cultured in DMEM^+^ alone (*p* < 0.05).

### 3.3. Experiment 2: Potential of CIMI to Reduce Damage Caused by DOXO on In Vitro Culture of Mouse Ovaries

The results showed that the presence of 0.3 μg/mL DOXO in culture medium significantly reduced the percentage of morphologically normal follicles. On the other hand, ovaries cultured in medium with 5 ng/mL CIMI maintained the percentage of morphologically normal follicles similar to those cultured in the DMEM^+^ alone. The presence of CIMI (5 ng/mL) in culture medium protected the ovaries against the deleterious effects caused by DOXO (0.3 μg/mL) (Figure 4). Figure 5A–F shows morphologically normal or degenerated follicles cultured in vitro.

The ovaries cultured with CIMI or both CIMI and DOXO showed a similar percentage of primordial and developing follicles, when compared to those cultured in DMEM^+^ alone (Figure 6A). However, ovaries cultured in presence of DOXO had reduced percentage of primordial when compared to control group (Figure 6B).

### 3.4. Immunohistochemical Localization of Active Caspase-3 in Mice Cultured Ovary

The immunohistochemical results showed that in the ovaries cultured with CIMI (5 ng/mL) alone or with both CIMI (5 ng/mL) and DOXO, caspase-3 staining was lower than that seen in ovaries cultured in cultured in DMEM^+^ alone or supplemented with DOXO (0.3 μg/mL) (Figure 7).

### 3.5. Evaluation of the Ovarian Extracellular Matrix after In Vitro Culture with CIMI and DOXO

After culture, no difference in the organization of collagen fibers in extracellular matrix was seen in ovaries cultured in the different treatments. Figure 8 shows regions containing collagen fibers after in vitro culture of mice ovaries.

### 3.6. Evaluation of Stromal Cells Density after In Vitro Culture of Mice Ovaries

Ovaries cultured with CIMI alone or in combination with DOXO maintained a well-preserved ovarian structure, similar to those cultured in control group (DMEM^+^). However, the presence of only DOXO in culture medium decreased the number of stromal cells (Figure 9). Figure 10 shows histological sections of mouse ovaries cultured in DMEM^+^ alone or supplemented with CIMI, DOXO or both CIMI and DOXO.

### 3.7. Viability Assessment of Follicles after Culture

Follicles isolated from ovaries cultured in presence of CIMI alone or both CIMI and DOXO showed follicles stained positively by calcein-AM, while follicles from ovaries cultured in presence of DOXO were positively stained for ethidium homodimer (Figure 11).

### 3.8. Ultrastructural Analysis after In Vitro Culture of Mice Ovaries

Ovaries cultured for 6 days in control medium supplemented with CIMI had well-preserved granulosa cells, increased mitochondrial activation with significant dilation of mitochondrial cristae (Figure 12E). Furthermore, it is possible to observe the presence of lipid inclusions in granulosa cells of ovaries cultured in control medium alone or supplemented with CIMI alone or associated with DOXO (Figure 12A,E,G). On the other hand, DOXO treatment showed few lipid inclusions (Figure 12C). Furthermore, the oocyte from the ovaries cultured in control medium alone or supplemented with isolated CIMI and DOXO showed intact zona pellucida, well-delimited nuclear membrane and poorly developed organelles as expected (Figure 12B,D,F). However, the oocyte from the ovaries cultured in a medium supplemented with CIMI and DOXO associated presented a zona pellucida with structural alterations, loss of oocyte shape, but a well-delimited nuclear membrane with poorly developed organelles as expected (Figure 12H).

### 3.9. Levels of mRNA for SOD, CAT and NRF2 in Cultured Ovaries

The results showed that ovaries cultured with CIMI alone had increased mRNA levels for SOD than those cultured in presence of DOXO. On the other hand, ovaries cultured with CIMI, DOXO or both CIMI and DOXO had levels of mRNA for SOD similar to those cultured in control medium (Figure 13A). The presence of both CIMI and DOXO significantly increased the levels of mRNA for CAT in cultured ovaries when compared to those cultured in medium containing DOXO (*p* < 0.05) (Figure 13B). In addition, it is possible to observe that DOXO reduced the mRNA expression for NRF2. On the other hand, ovaries cultured with both CIMI and DOXO showed a lower expression for NRF2 than those culture with only DOXO (*p* < 0.05) (Figure 13C).

## 4. Discussion

This study demonstrates for the first time that CIMI extract attenuates DOXO-induced damage on follicular morphology. The CIMI extract with its high content of antioxidants, such as methyl caffeine, ferulic acid, isoferulic acid, fucinolic acid, cimicifugic acid A, B and F, cimiracemate A, and cyimiracemate B, is able to protect ovaries against ROS produced in response to DOXO [24,25]. Furthermore, actein, a compound isolated from CIMI, reduced cytotoxicity in osteoblastic cells by increasing the activity of glyoxalase I and levels of reduced glutathione (GSH) and the transcription factor nuclear factor erythroid 2-related factor 2 (Nrf2) [26]. DOXO uses the oxidative stress pathway and apoptotic signaling to induce ovarian damage, oxidative stress is positively correlated with lipid peroxidation and negatively correlated with increased activity of antioxidant enzymes [27]. In the present study, compared to ovaries cultured with DOXO, CIMI increase the levels of mRNA for SOD and CAT. SOD provides high protection against tissue oxidative damage by dismutation of superoxide radicals. SOD and non-enzymatic antioxidants neutralized ROS by dismutating O_2_^−^ into H_2_O_2_ and oxygen, and then CAT reduces H_2_O_2_ to water and oxygen, promoting ovarian protection [28]. This enzymatic antioxidant defense contributes to remove harmful products and certainly help to keep ROS levels within physiological limits, which protects ovarian follicles and stromal cells from degenerative events.

The presence of DOXO in culture medium reduced density of stromal cells in mice ovaries cultured in vitro. The follicular development is strongly influenced by ovarian stromal cells, which supports tissue and play several important roles. In general, fibroblasts from ovarian stromal tissue secrete proteins of ECM, such as collagen, for cellular support, structure and repair. Furthermore, it promotes the formation of follicular capillaries, which provide nutrients for follicular growth and development [29]. Our findings are consistent with reports from previous studies, in which DOXO causes damage to ovarian microvasculature and increases the production of ROS, triggering cell death by apoptosis which induces the loss of stromal cells [30,31]. Interestingly, we show that CIMI also contribute to keep stromal cell density similar to control group.

In this study, ovaries cultured with CIMI had lower staining for activated caspase 3. Caspase-3 has already been shown to be a marker of apoptosis in mammalian cells and initiates the apoptotic cascade by activating other caspases [32]. Compounds with antioxidant or anti-inflammatory properties can protect ovarian tissue by reducing the level of caspase-3 after chemotherapy-induced toxicity [33]. Corroborating the findings of this study, Wang et al. [5] had already demonstrated the involvement of caspase-3 in DOXO-induced damage pathways in follicular and oocyte death.

The CIMI extract did not influence the distribution of ECM in cultured ovaries. Several studies have shown that the presence of collagen fibers in the cortical stroma after chemotherapy is a characteristic sign of cell death and healing of ovarian tissue [34]. However, due to the short cultured period of six days, increase in follicular degeneration was not associated with changes in ECM. It is important to highlight that CIMI has the ability to inhibit collagenases [35,36,37] that can help to preserve ovarian ECM in longer culture periods. Furthermore, the compounds present in the rhizome of species of the genus *Cimicifuga* are able to prevent collagen degradation through the inhibition of collagen-like matrix metalloproteinases [35].

Previous studies report that DOXO accumulates in the developing follicles, i.e., secondary, and tertiary follicles accumulate more DOXO than primordial and primary follicles in mice ovaries [38]. Exposure to DOXO results in loss of both primordial and growing follicles, mainly affecting mitotically active granulosa cells, and leading to a reduction in ovulation rates [30,39,40,41]. In the present study, ovaries cultured in the presence of DOXO reduced the percentage of primordial follicles, while the presence of CIMI alone kept the percentage of primordial and developing follicles. To explain this fact, Wang et al. [5] demonstrated that DOXO causes overactivation of primordial follicles and consequently increases the number of developing follicles. Despite the presence of growing follicles inhibits the recruitment of primordial follicles, the chemotherapy drugs have a cytotoxic effect on growing follicles and the increasing loss of this follicular population leads to an increase in the activation of primordial follicles, and thus, the loss of follicular reserve [5]. Thus, increasing cellular enzymatic defense against oxidative stress will protect the ovarian follicle reserve and help to maintain fertility in patients undergoing chemotherapy.

## 5. Conclusions

The presence of 5 ng/mL CIMI in culture medium protects mice ovaries against DOXO-induced toxicity. Taken together, CIMI preserves ovarian structure and reduces expression of activated caspase-3. Further studies are needed in order to assess the in vivo therapeutic effects of CIMI on female gametes, especially when they are exposed to substances that are harmful to reproductive health.

## Figures and Tables

**Figure 1 animals-13-00018-f001:**
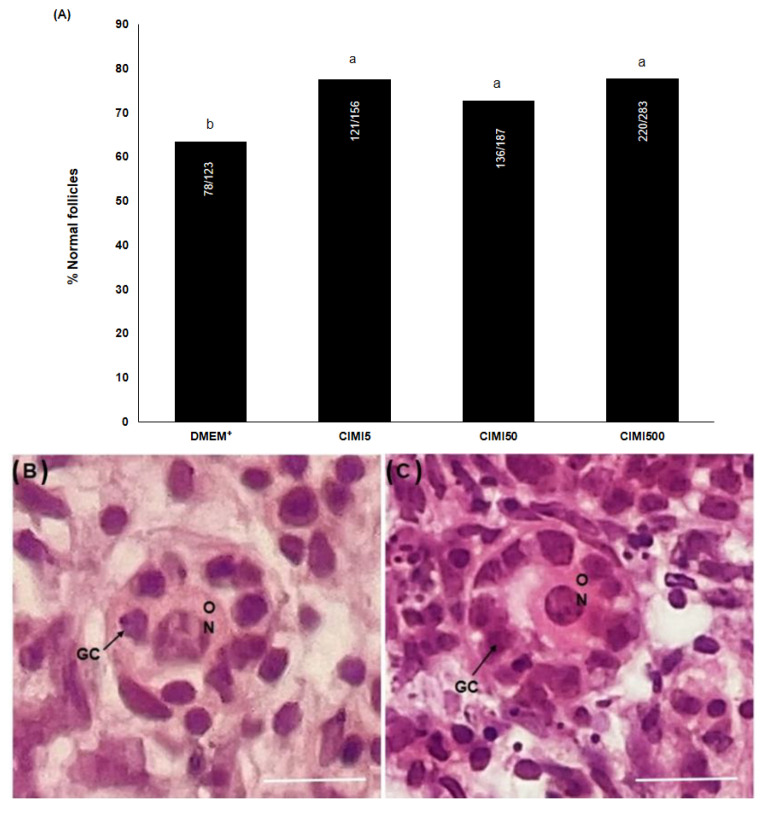
The percentage of normal follicles in ovaries after 6 days of culture in DMEM^+^ alone or supplemented with CIMI (5 ng/mL, 50 ng/mL or 500 ng/mL) (**A**). Histological sections of degenerated follicle cultured in DMEM^+^ (**B**). Normal preantral follicles cultured in the presence of CIMI (5 ng/mL) (**C**). The percentage of normal follicles was compared by chi-square test. a and b different lowercase letters indicate statistically significant differences between treatments (*p* < 0.05). Numbers of follicles evaluated are shown within each column. a,b: different lowercase letters indicate statistically significant differences between treatments (*p* < 0.05). O: oocyte, GC: granulosa cells; N: nucleus. Scale bar: 100 μm (400×).

**Figure 2 animals-13-00018-f002:**
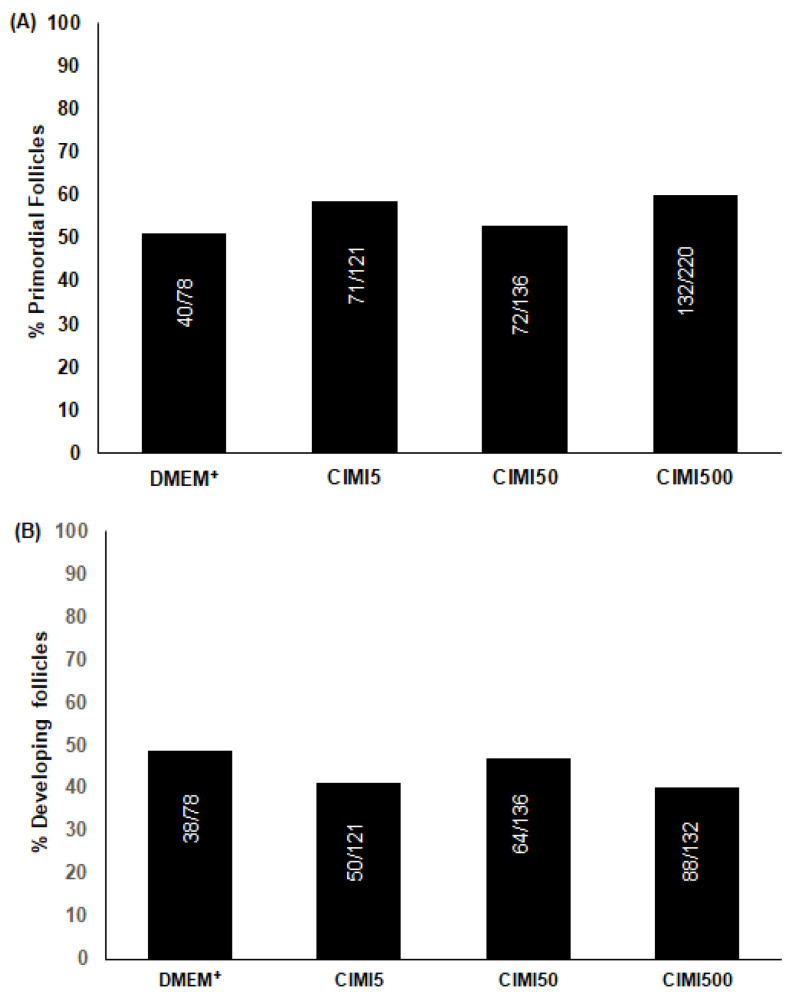
The percentage (mean ± SE) of primordial follicles (**A**) and developing follicles (**B**) in ovaries cultured for 6 days in DMEM^+^ alone or with CIMI at different concentrations (5 ng/mL, 50 ng/mL or 500 ng/mL). The percentage of primordial and developing follicles was compared by chi-square test and no significant differences were observed between treatments (*p* > 0.05). Numbers of follicles evaluated are shown within each column.

**Figure 3 animals-13-00018-f003:**
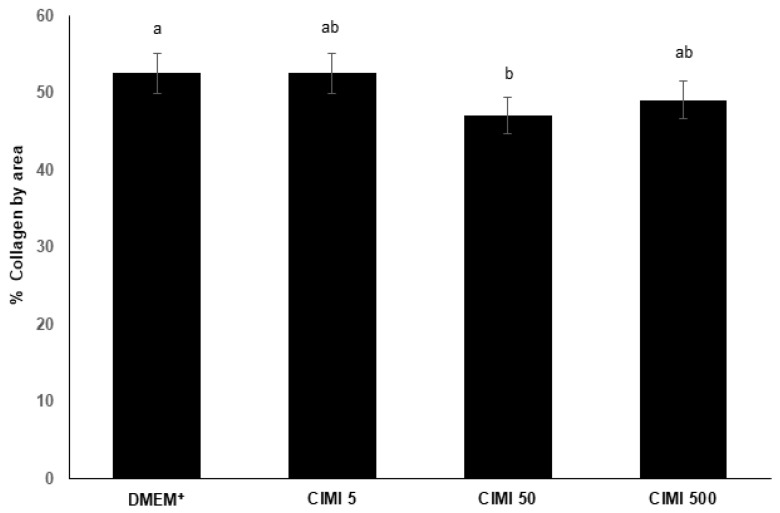
The levels of collagen fibers (mean ± SD) in the mouse ovaries (n = 9 ovaries per treatment) cultured for 6 days in DMEM^+^ alone or supplemented with CIMI (5 ng/mL, 50 ng/mL or 500 ng/mL). Collagen fiber distribution was analyzed by the Kruskal-Wallis test, followed by Dunn’s comparison. a and b different lowercase letters indicate statistically significant differences between treatments (*p* < 0.05).

**Figure 4 animals-13-00018-f004:**
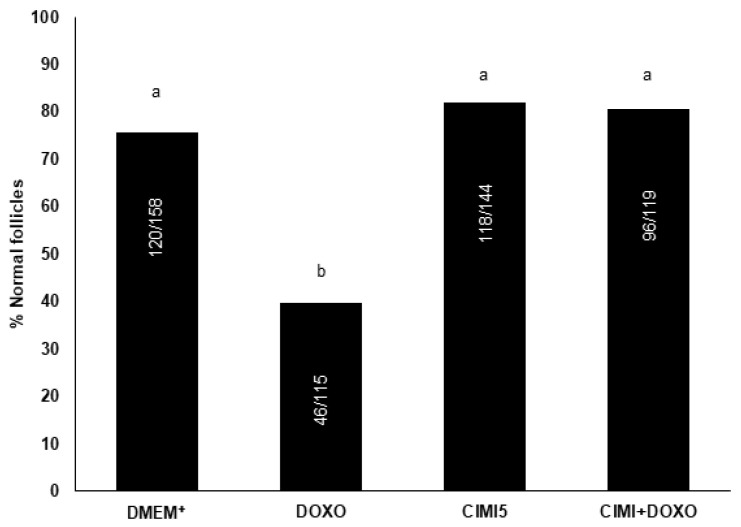
The percentage (mean ± SE) of normal follicles in ovaries cultured in DMEM^+^ alone or supplemented with DOXO (0.3 μg/mL), CIMI (5 ng/mL) and CIMI+DOXO (5 ng/mL + 0.3 μg/mL). The percentage of normal follicles was evaluated using the chi-square test. a and b different lowercase letters indicate statistically significant differences between treatments (*p* < 0.05). Numbers of follicles evaluated are shown within each column.

**Figure 5 animals-13-00018-f005:**
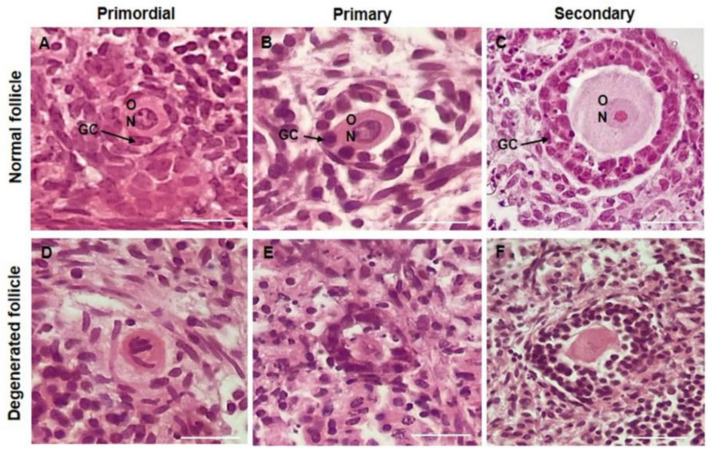
Representative images of ovarian sections of mouse cultured in vitro for 6 days showing a morphologically normal (**A**–**C**) and degenerated (**D**–**F**) follicles from different stages of development stained with haematoxylin and eosin. (**A**) Normal and (**D**) degenerated primordial follicles; (**B**) Normal and (**E**) degenerated primary follicles; (**C**) Normal and (**F**) degenerated secondary folliclse. Granulosa cells (GC); Oocyte (O); Oocyte nucleus (N). (400×, Scale bar: 100 μm).

**Figure 6 animals-13-00018-f006:**
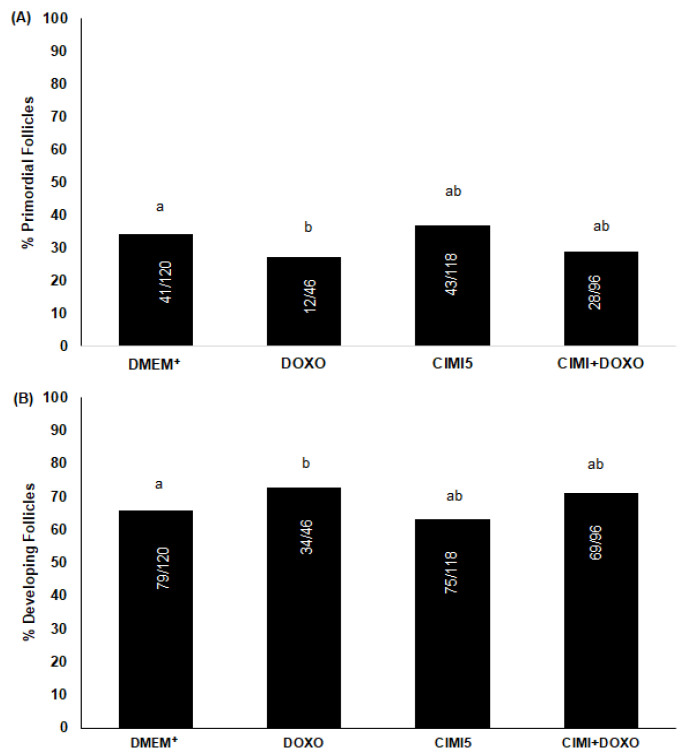
The percentage (mean ± SE) of primordial (**A**) and developing (**B**) follicles in ovaries cultured in DMEM^+^ alone or supplemented with DOXO (0.3 μg/mL), CIMI (5 ng/mL) and both CIMI and DOXO (5 ng/mL + 0.3 μg/mL). The percentage of primordial and developing follicles was evaluated using the chi-square test. a and b different lowercase letters indicate statistically significant differences between treatments (*p* < 0.05). Numbers of follicles evaluated are shown within each column.

**Figure 7 animals-13-00018-f007:**
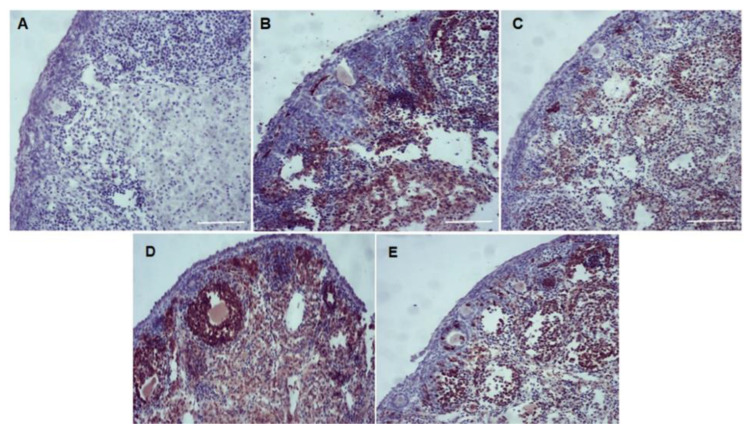
Immunohistochemical staining for caspase-3 in mice ovaries before (**A**) and after culture in vitro for 6 days in the presence or absence of CIMI or DOXO (**B**–**E**): (**A**) negative control; (**B**) control group (DMEM^+^); (**C**) CIMI (5 ng/mL); (**D**) DOXO (0.3 μg/mL); (**E**) and CIMI + DOXO. (100×—Scale bar: 100 µm).

**Figure 8 animals-13-00018-f008:**
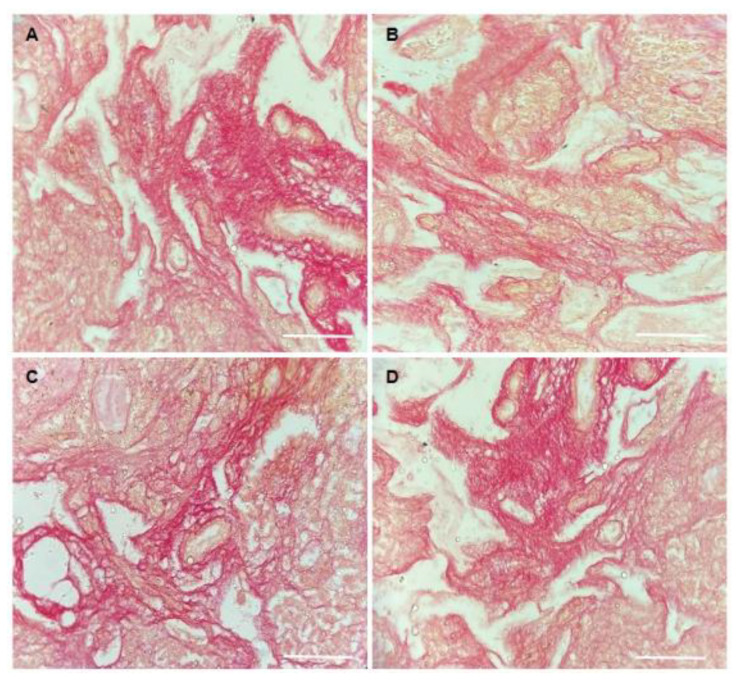
Representative images of collagen fibers labeled by Picrosirius red: (**A**) control group (DMEM^+^); (**B**) DOXO (0.3 μg/mL); (**C**) CIMI (5 ng/mL) and (**D**) CIMI (5 ng/mL) + DOXO (0.3 μg/mL). Scale bar: 100 μm (400×).

**Figure 9 animals-13-00018-f009:**
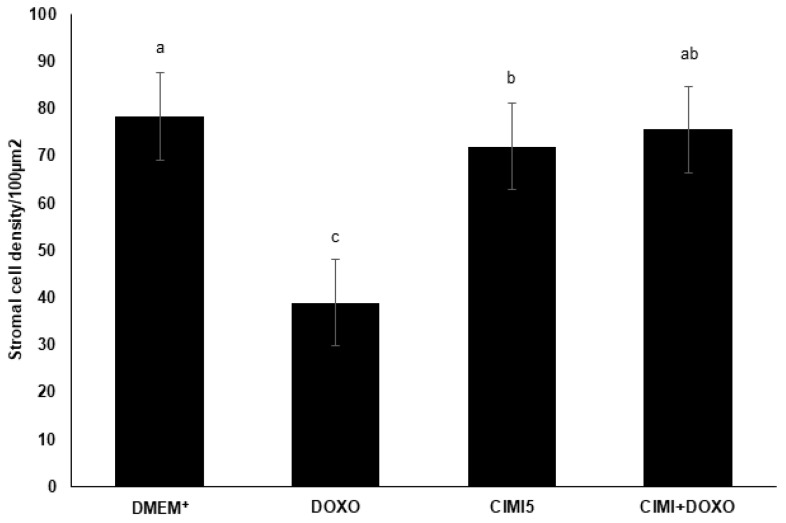
Stromal cell density on ovarian tissues (n = 09 ovaries per treatment) after culturing in DMEM^+^ alone or with DOXO (0.3 μg/mL), CIMI (5 ng/mL) and both CIMI and DOXO. Stromal density was analyzed using the Kruskal-Wallis test, followed by Dunn’s comparison. a, b, c different lowercase letters indicate statistically significant differences between treatments (*p* < 0.05).

**Figure 10 animals-13-00018-f010:**
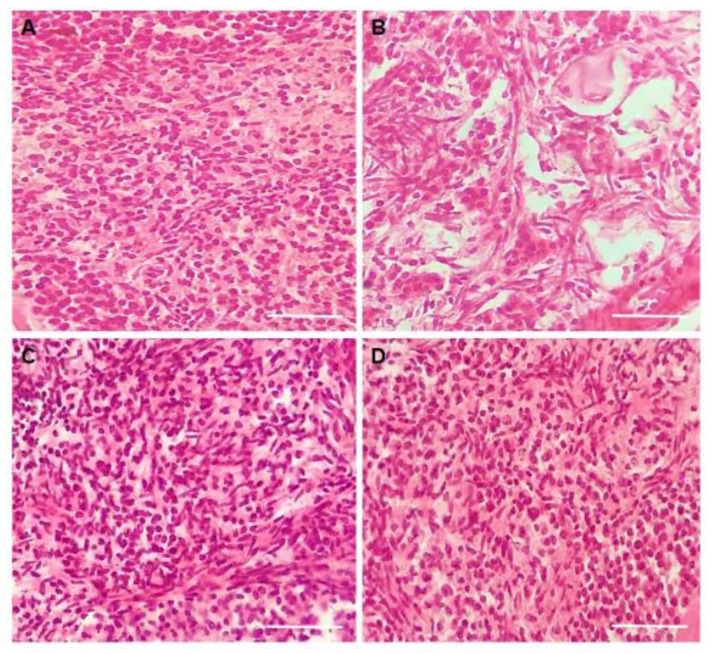
Representative images of ovarian stromal density after 6 days of culture: (**A**) control group (DMEM^+^); (**B**) DOXO (0.3 μg/mL); (**C**) CIMI (5 ng/mL) and (**D**) CIMI (5 ng/mL) + DOXO (0.3 μg/mL). Scale bar: 100 μm (400×).

**Figure 11 animals-13-00018-f011:**
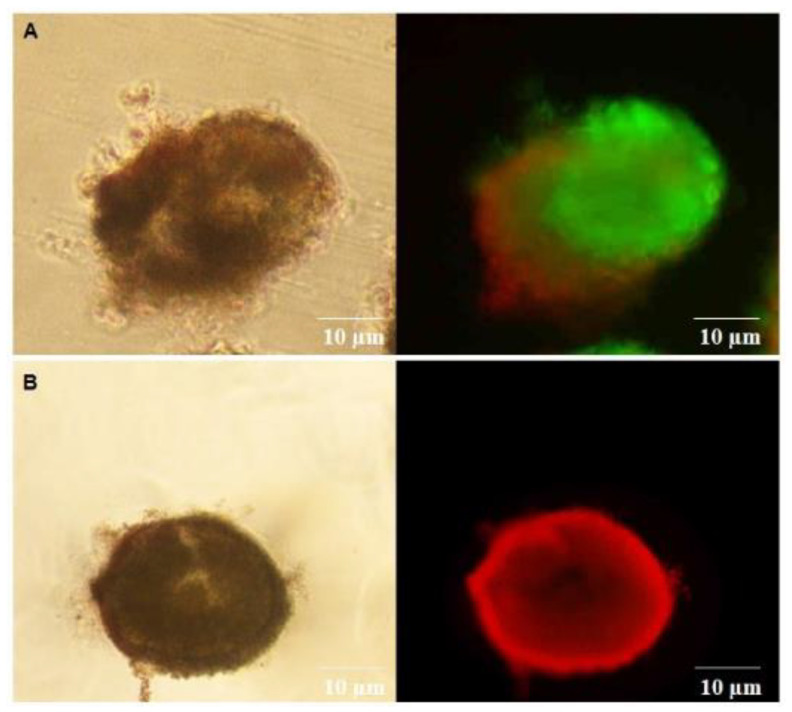
The valuation of follicular viability after staining with calcein-AM (green- viable follicle) and ethidium homodimer-1 (red-non-viable follicle). (**A**) follicle cultured with both CIMI and DOXO; (**B**) follicle cultured with DOXO. Scale bars represent 10 μm.

**Figure 12 animals-13-00018-f012:**
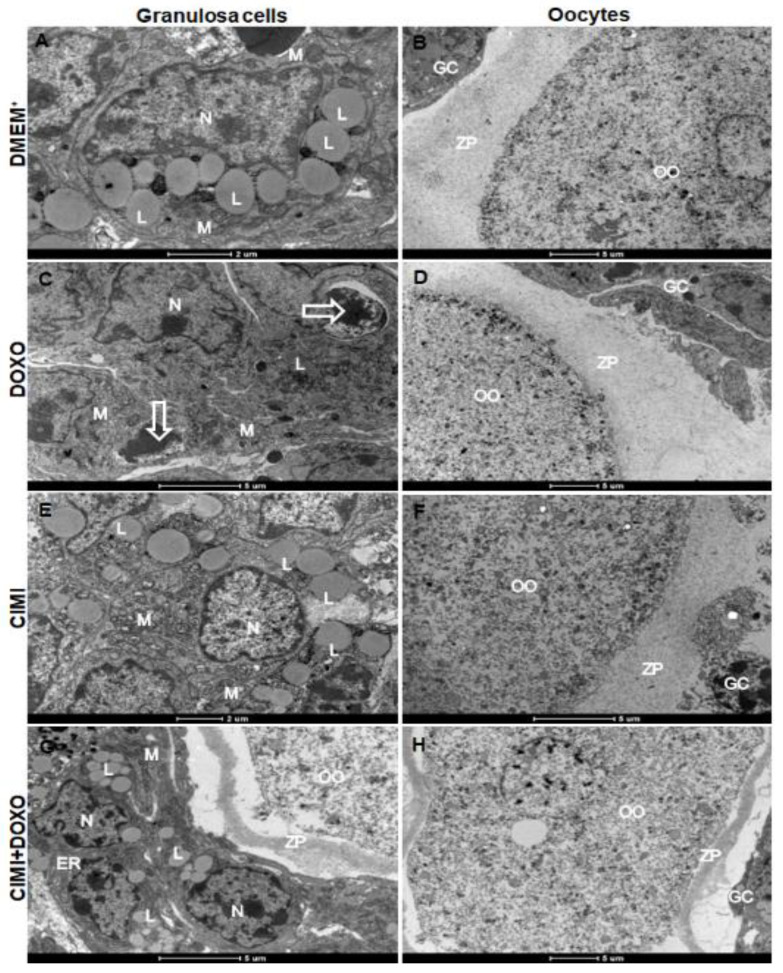
Representative micrographs of mice ovaries cultured for 6 days: (**A**,**B**) control group (DMEM^+^); (**C**,**D**) DOXO (0.3 μg/mL); (**E**,**F**) CIMI (5 ng/mL) and (**G**,**H**) CIMI (5 ng/mL) + DOXO (0.3 μg/mL). M, mitochondria; N, nucleus; ZP, zona pellucida, L, lipid; OO, ooplasm; GC, granulosa cells; ER, endoplasmic reticulum.

**Figure 13 animals-13-00018-f013:**
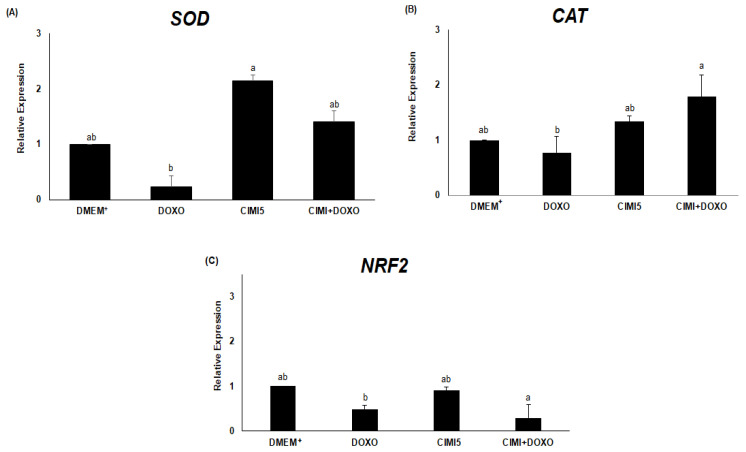
The levels of mRNA for (**A**) SOD, (**B**) CAT and (**C**) NRF2 in ovaries (n = 04 ovaries per treatment) after cultured in control group (DMEM^+^), DOXO (0.3 μg/mL), CIMI (5 ng/mL) and CIMI+DOXO (5 ng/mL + 0.3 μg/mL). Levels of mRNA for SOD, CAT and NRF2 were analyzed by the Kruskal-Wallis test, followed by the Dunn comparison. a, b different lowercase letters indicate significant difference between treatments (*p* < 0.05).

**Table 1 animals-13-00018-t001:** The primers pairs used for real-time PCR.

Target Gene	Primer Sequence (5′ → 3′)	Forward (F) Reverse (R)	GenBank Accession No.
GAPDH	GAACGGATTTGGCCGTATTG GTGAGTGGAGTCATACTGGAAC	F R	GU214026.1
SOD	CTCGTCTTGCTCTCTCTGGTC CTTGCCTTCTGCTCGAAGTG	F R	NM_011434.2
CAT	CCAATGGCAATTACCCGTCC CCTTGTGAGGCCAAACCTTG	F R	NM_009804.2
NRF2	TTGCCCTAGCCTTTTCTCCG CTAGGAGATAGCCTGCTCGC	F R	NM_010902.4

## Data Availability

The data presented in this study are openly available in the repository Federal University of Ceará (https://repositorio.ufc.br/).

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
