# Peer review of "Protective Effect of Cimicifuga racemosa (L.) Nutt Extract on Oocyte and Follicle Toxicity Induced by Doxorubicin during In Vitro Culture of Mice Ovaries"

_animals, 2022, doi:10.3390/ani13010018_

Round 1

Reviewer 1 Report

Comments to the manuscript “Protective effect of Cimicifuga racemosa (L.) Nutt extract on oocyte and follicle toxicity induced by doxorubicin during in 3 vitro culture of mice ovaries”

The manuscript is original, since it was demonstrated for the first time that CIMI extract attenuates DOXO-induced damage on follicular morphology and density of ovarian stromal cells in mice ovaries. I just have the following minor comments.

Line 28: Please indicate that Higher staining intensity for activated caspase-3…

Line 85: Please indicate the implications of the reported reduction in the expression of Ki-67 in granulosa, theca and stromal cells.

Line 120: Please indicate if ovaries were isolated from newborn mice and placed on Millicell-PC membrane inserts.

Line 126: Please indicate how these CIMI concentrations were selected for the study.

Line 193: Please correct “The, the”

Line 225: Please note that it is not mentioned how the Retrotranscription was carried out.

Table 1: Please consider replacing Sense and Antisense for Forward and Reverse.

Figure 1, 2, 4 and 6: Please note that the SD bars are missing

Figure 6: Please explain in the discussion section why the number of developing follicles is higher in the DOXO group than in the DMEM+ group

Author Response

The manuscript is original, since it was demonstrated for the first time that CIMI extract attenuates DOXO-induced damage on follicular morphology and density of ovarian stromal cells in mice ovaries. I just have the following minor comments.

  1. Line 28: Please indicate that Higher staining intensity for activated caspase-3…

RESPONSE: We thank the reviewer for the comment. The phrase was revised and rewritten. L43; P1: “A higher staining intensity for caspase-3 was seen in ovaries cultured in (…)”

  1. Line 85: Please indicate the implications of the reported reduction in the expression of Ki-67 in granulosa, theca and stromal cells.

RESPONSE: The text was revised at L86-88; P2: In vivo, CIMI extract improved antioxidant status, hormonal and lipid profile, glucose levels and liver functions in rats with polycystic ovary syndrome (PCOS), besides inducing the expression of Ki-67 in granulosa, theca and stromal cells. Expression of Ki67 is an indicator of cellular proliferation and its increase is indicative of high androgen production in ovaries of patients with PCOS [13].”

  1. Line 120: Please indicate if ovaries were isolated from newborn mice and placed on Millicell-PC membrane inserts.

RESPONSE: This information was better described in the text and marked in yellow color. L123; P3: “The ovaries (n=36) from mice (n=18) with 18 g and/or 2 months old were cultured according to the protocol described by O’Brien [16]. In experiment 1, the ovaries were collected using surgical scissors and 28 G needles and cultured individually in 24-well plates.”

  1. Line 126: Please indicate how these CIMI concentrations were selected for the study.

RESPONSE: Experiment 1 was conducted to define the better concentration of CIMI to be used in association with DOX. In this way, a dose-response curve (5, 50 or 500 ng/ml) of CIMI was established to define the better concentration. We added this sentence in the text – L130-131; P3: “This dose-response curve was established to define the better concentration of CIMI to be used in experiment 2.”

  1. Line 193: Please correct “The, the”

RESPONSE: The sentence has been revised and rewritten. L199; P5: “the” was removed.

  1. Line 225: Please note that it is not mentioned how the Retrotranscription was carried out.

RESPONSE: We thanks for de comment. This information was added in the text. L230-236; P5: “Reverse transcription was performed in a total volume of 20 μl composed of 10 μl of sample containing 1 mg of RNA, 4 μl reverse transcriptase buffer (Invitrogen), eight units RNAsin, 150 units of reverse transcriptase Superscript III, 0.036 U random primers, 10 mM dithiothreitol and 0.5 mM of each dNTP (Invitrogen). The mixture was incubated at 42.1 â—¦C for 1 h, subsequently at 80 â—¦C for 5 min, and finally stored at − 20 â—¦C. The negative control was prepared under the same conditions but without the addition of reverse transcriptase.”

  1. Table 1: Please consider replacing Sense and Antisense for Forward and Reverse.

RESPONSE: The text was revised and rewrite. The Forward (F) and Reverse (R) words were added in the table 1 and marked in yellow color.

  1. Figure 1, 2, 4 and 6: Please note that the SD bars are missing

RESPONSE: We thank the reviewer for the comment. In Figure 1, follicles counted in all ovaries were pooled and then used to calculate the percentages of normal follicles. The number of normal follicles / total are within each column. It is not the mean percentage; thus we cannot calculate SD values. The same type of data is shown in figures 2, 4 and 6. In the legends of each figure we added the following information “Numbers of follicles evaluated are shown within each column”.

  1. Figure 6: Please explain in the discussion section why the number of developing follicles is higher in the DOXO group than in the DMEM+ group

RESPONSE: The text was revised and this information was added. L4559-457; P16: “It was reported that DOXO causes overactivation of primordial follicles and consequently increases the number of developing follicles [5].

Reviewer 2 Report

I have revised the manuscript entitled “Protective effect of Cimicifuga racemosa (L.) Nutt extract on oocyte and follicle toxicity induced by doxorubicin during in vitro culture of mice ovaries" by de Assis et al. (animals-2070189). This paper is an ambitious exploration of the role of Cimicifuga racemosa (L.) Nutt extract (CIMI) in preventing ovarian damage caused by the chemotherapeutic drug, doxorubicin (DOXO), in cultured mouse ovary. Overall, the Authors revealed that the presence of 5 ng/ml CIMI in culture medium protects mouse ovaries against DOXO-induced toxicity. The paper deals with an interesting subject, and contains a lot of novel data that will contribute to the field of reproductive biology.  The experiment is appropriately designed and the methodology is adequate for accomplishing the specific objectives. However, it would be beneficial to perform at least TUNEL analysis to assess the anti-apoptotic impact of CIMI on ovarian follicles. Discussion section requires huge improvements. Discussion should report to the readers the most important findings of this work and to stress the correctness of the hypothesis. In the present form, Discussion represents rather the comparison of the obtained results with results obtained previously. Sometimes it is written in the style of a review paper, and it should be corrected. Please, stress the meanings of your own research and expand description of the meanings of your own research, especially in relation to woman. Each paragraph in Discussion section should end with a concluding statement. Additional comments that should be addressed are defined as follows:

Line 28: if active form of Caspase 3 has been examined? Please, clarify.

Line 37-38 and line 97: what does “extracellular matrix configuration” mean? Authors examined only percentage of collagen fibers and their organization.

Line 38: how the viability of the ovaries have been examined? Probably it should be “viability of oocytes”, please clarify.

Lines 117-118: It should be clearly stated what was the phase of estrous cycle of the examined animals. How many animals have been utilized for each experiment, “n” number should represent the number of animals, not number of ovaries, unless only one ovary per animal has been utilized. Please clearly state what was “n” number of the study. The Figure legends would also benefit from inclusion of “n” and type of Statistical test.

Lines 129-131: Please clearly state what were the criteria for choosing the “better concentration”of CIMI.

Line 137-145: How many sections from each ovary/animal have been examined?

“Overall, 130 follicles were evaluated for each treatment” – it is per each ovary or per each animal? Please state it clearly.

Line 146-149 Again, how many sections from each ovary/animal have been examined to evaluate cellular density in the ovaries? What is the “n” number?

Line 169: Again, the “n” number refers to number of animals?

Line 175: The immunohistochemistry is not a quantitative method. On the photometric side 1) amplification with avidin-biotin invalidates the stoichiometry of the reaction (staining intensity does not correspond to the amount of antigen); 2) counterstaining cannot be used because the absorption spectrum of hematoxylin partially overlaps with that one of the diaminobenzidine final product. The protein abundance should be evaluted by western blot. IHC is a good method for protein localization. In my opinion, the better way to show the apoptotic cells within tissue is TUNEL method.

Line 201: Please provide the catalog number of caspase 3 antibody. If this antibody detect only active form of caspase 3?

Lines 205-206: Replacing primary antibody with TBS-Tween solution is not an appropriate negative control. The respective non-immune rabbit IgG should be used instead.

Line 221: Again, what was the “n” number of the experiment?

Line 303: Only healthy primordial and primary follicles have been examined?

Line 307: Since IHC is not a quantitative method, the subheading should be changed

Line 309: Please see the comment referring to line 175. If the antibody detects only active form of caspase 3? If yes, it would be beneficial to confirm it using Western blot method.   

Minor:

Line 34: it is: ”supplemented 5”, it should be: “supplemented with 5”

Author Response

Reviewer #2 - Comments to the manuscript “Protective effect of Cimicifuga racemosa (L.) Nutt extract on oocyte and follicle toxicity induced by doxorubicin during in vitro culture of mice ovaries”

  1. “(…) However, it would be beneficial to perform at least TUNEL analysis to assess the anti-apoptotic impact of CIMI on ovarian follicles. (…)”

RESPONSE: We thank the reviewer for the comment. In the present study, we investigated the expression of caspase-3 in the mouse ovary after culture and determined whether active caspase-3 was present. The expression and activation of caspase-3 induce apoptosis in a number of different types of cell and its expression has been found in the ovary of hens, quails, rats and mice and humans. The localization of the active form of caspase-3 not only implicates it in follicular atresia but also serves as a useful and reliable marker for the identification of apoptotic cells. Activated Caspase 3 marks only apoptotic follicles, while TUNEL evaluation marks DNA fragmentation, which can occur due to other types of cell death.

Fenwick MA, Hurst PR. Immunohistochemical localization of active caspase-3 in the mouse ovary: growth and atresia of small follicles. Reproduction. 2002 Nov;124(5):659-65. doi: 10.1530/rep.0.1240659. PMID: 12417004.

Glamoclija V, Vilović K, Saraga-Babić M, Baranović A, Sapunar D. Apoptosis and active caspase-3 expression in human granulosa cells. Fertil Steril. 2005 Feb;83(2):426-31. doi: 10.1016/j.fertnstert.2004.06.075. PMID: 15705385.

  1. Discussion section requires huge improvements. Discussion should report to the readers the most important findings of this work and to stress the correctness of the hypothesis. In the present form, Discussion represents rather the comparison of the obtained results with results obtained previously. Sometimes it is written in the style of a review paper, and it should be corrected. Please, stress the meanings of your own research and expand description of the meanings of your own research, especially in relation to woman. Each paragraph in Discussion section should end with a concluding statement.

RESPONSE: We have improved the discussion and all correction are marked in yellow color.

  1. Line 28: if active form of Caspase 3 has been examined? Please, clarify.

RESPONSE: In the present study, the antibody used in the caspase 3 IHC reaction is specific for the cleaved (active) form of the caspase protein (ant-cleaved caspase-3 ab49822, abcam).

  1. Line 37-38 and line 97: what does “extracellular matrix configuration” mean? Authors examined only percentage of collagen fibers and their organization.

RESPONSE: Collagen is the most abundant component of the extracellular matrix. Therefore, the modification in the amount and distribution of collagen can provide relevant information of the ovarian structure was after the culture period. The Picrosirus red stain the collagen fiber and enable us to evaluated their organization. 

  1. Line 38: how the viability of the ovaries have been examined? Probably it should be “viability of oocytes”, please clarify.

RESPONSE: In this study the viability of the ovarian follicles after culturing ovarian tissue was performed by fluorescence microscopy. After 6 days of culture, the follicles were isolated from the ovaries and viability was performed using fluorescent probes by calcein-AM (live cells) and homodimer ethidium-1 (dead cells). This information was added in the abstract and marked in yellow color. L38-39; P1: “viability by fluorescence microscopy from the cultured ovarian follicles.”

  1. Lines 117-118: It should be clearly stated what was the phase of estrous cycle of the examined animals. How many animals have been utilized for each experiment, “n” number should represent the number of animals, not number of ovaries, unless only one ovary per animal has been utilized. Please clearly state what was “n” number of the study. The Figure legends would also benefit from inclusion of “n” and type of Statistical test.

RESPONSE: The animals were randomly assigned to the treatments and the estrous cycle was performed in order to assess whether the animals had a regular cycle. The “n” refers to the number of ovaries used for each experiment and is indicated in the methodology of each experiment. The legend of the figures has been revised and rewritten.

L267-272; P7: “Figure 1: …The percentage of normal follicles was evaluated using the chi-square test. a and b different lowercase letters indicate statistically significant differences between treatments (P <0.05).”

L279-283; P7-8: “Figure 2: …The percentage of primordial and developing follicles was evaluated using the chi-square test…”

L290-293; P8: “Figure 3: …Collagen fiber distribution was analyzed by the Kruskal-Wallis test, followed by Dunn's comparison…”

L304-307; P9: “Figure 4: …The percentage of normal follicles was evaluated using the chi-square test…”

L319-323; P10: “Figure 6: …The percentage of primordial and developing follicles was evaluated using the chi-square test…”

L352-355; P12: “Figure 10: …Stromal density was analyzed using the Kruskal-Wallis test, followed by Dunn's comparison. a, b and c different lowercase letters indicate statistically significant differences between treatments (P <0.05).

L398-402; P15: “Figure 14: …Levels of mRNA for SOD, CAT and NRF2 were analyzed by the Kruskal-Wallis test, followed by the Dunn comparison…”

  1. Lines 129-131: Please clearly state what were the criteria for choosing the “better concentration”of CIMI.

RESPONSE: The experiment 1 was conducted to define the better concentration of CIMI to be used in association with DOX. On this way, a dose-response curve (5, 50 or 500 ng/ml) of CIMI was established to define the better concentration.  5 ng/mL CIMI did not show damage to the ovaries, and maintains the higher percentage of normal follicles compared to control group (DMEM+).

  1. Line 137-145: How many sections from each ovary/animal have been examined?

“Overall, 130 follicles were evaluated for each treatment” – it is per each ovary or per each animal? Please state it clearly.

RESPONSE: Dear reviewer, follicles were counted in every third section of the ovary and they were classified as primordial, primary and secondary follicles based on their morphological appearance as detailed in the literature. Furthermore, the oocyte nucleus was used as a defining characteristic, to avoid overestimation.

  1. Line 146-149 Again, how many sections from each ovary/animal have been examined to evaluate cellular density in the ovaries? What is the “n” number?

RESPONSE: Dear reviewer, 5 ovaries from each treatment were destined for histological analysis. For stromal density, at least 3 sections of each ovary were analyzed and 5 fields from each section were used for this analysis.

  1. Line 169: Again, the “n” number refers to number of animals?

RESPONSE: Dear reviewer, the “n” refers to the number of ovaries used for each experiment.

  1. Line 175: The immunohistochemistry is not a quantitative method. On the photometric side 1) amplification with avidin-biotin invalidates the stoichiometry of the reaction (staining intensity does not correspond to the amount of antigen); 2) counterstaining cannot be used because the absorption spectrum of hematoxylin partially overlaps with that one of the diaminobenzidine final product. The protein abundance should be evaluated by western blot. IHC is a good method for protein localization. In my opinion, the better way to show the apoptotic cells within tissue is TUNEL method.

RESPONSE: We thank the reviewer for the comment. We have removed the data about qualitative analysis of caspase-3 expression. Only qualitative data was kept in the manuscript. Western blot and TUNEL methods will certainly be considered in future studies.

  1. Line 201: Please provide the catalog number of caspase 3 antibody. If this antibody detect only active form of caspase 3?

RESPONSE: This information was added in the text: L207; P5: (Anti-Caspase-3 antibody (ab49822), ABCAM).

  1. Lines 205-206: Replacing primary antibody with TBS-Tween solution is not an appropriate negative control. The respective non-immune rabbit IgG should be used instead.

RESPONSE: There was a mistake in the description, we revised it since normal rabbit IgG was used in the negative control. (L. 213- 214)

  1. Line 221: Again, what was the “n” number of the experiment?

RESPONSE: Dear reviewer, the “n” refers to the number of ovaries used for each experiment.

  1. Line 303: Only healthy primordial and primary follicles have been examined?

RESPONSE: Yes, for the percentage of primordial follicles (Figure 6A) normal follicles were evaluated and for the percentage of developing follicles (Figure 6B) normal primary and secondary follicles were evaluated. Degenerated follicles were considered in this analysis.

Line 307: Since IHC is not a quantitative method, the subheading should be changed

RESPONSE: The sentence has been revised and rewritten. L324; P10: “Immunohistochemical localization of active caspase-3 in mice cultured ovary”.

Line 309: Please see the comment referring to line 175. If the antibody detects only active form of caspase 3? If yes, it would be beneficial to confirm it using Western blot method.     

RESPONSE: In the present study we used Anti-Caspase-3 antibody (ab49822)- ABCAM). Previous studies have already used this same antibody to localize active caspase-3

Fenwick MA, Hurst PR. Immunohistochemical localization of active caspase-3 in the mouse ovary: growth and atresia of small follicles. Reproduction. 2002 Nov;124(5):659-65. doi: 10.1530/rep.0.1240659. PMID: 12417004.

Glamoclija V, Vilović K, Saraga-Babić M, Baranović A, Sapunar D. Apoptosis and active caspase-3 expression in human granulosa cells. Fertil Steril. 2005 Feb;83(2):426-31. doi: 10.1016/j.fertnstert.2004.06.075. PMID: 15705385.

Minor:

Line 34: it is: “supplemented 5”, it should be: “supplemented with 5”

RESPONSE: The text was revised and rewrite. L34; P1:supplemented 5” was replaced for “supplemented with 5”.  

Round 2

Reviewer 2 Report

Revision improved this manuscript substantially. However, as I still have a few  comments, the manuscript requires revision before the publication in Animals. The specific comments that should be addressed are defined as follows:

  1. Apoptotic cell analysis.

According to data sheet anti-Caspase-3 antibody (ab49822) reacts only with humans. More details regarding the validation of the antibody used for IHC is required to confirm that positive reaction refers only to cleaved caspase 3 form e.g. Western blot is required (or a reference to previous validation for mouse tissue), this includes assessment of non-specific binding for the Western blot and inclusion of a positive and negative control tissue.

  1. Figure 3, 10 and 14: please add “n” number in figure description.

Author Response

Comments:

Reviewer #2 – Comments to the manuscript “Protective effect of Cimicifuga racemosa (L.) Nutt extract on oocyte and follicle toxicity induced by doxorubicin during in vitro culture of mice ovaries”

  1. Apoptotic cell analysis.

According to data sheet anti-Caspase-3 antibody (ab49822) reacts only with humans. More details regarding the validation of the antibody used for IHC is required to confirm that positive reaction refers only to cleaved caspase 3 form e.g. Western blot is required (or a reference to previous validation for mouse tissue), this includes assessment of non-specific binding for the Western blot and inclusion of a positive and negative control tissue.

RESPONSE: Dear reviewer, this anti-Caspase-3 antibody (ab49822) has already been used in another experiment with mouse. This information was added in the text: L209-210; P5: “…The specificity of this antibody has already been evaluated in previous studies with mice [22].”.

  1. Figure 3, 10 and 14: please add “n” number in figure description.

RESPONSE: Thanks to the reviewer for the comment. The "n" was added in the description of the figures 3, 10 and 14.